# Poly(ethylene-co-vinyl alcohol) Electrospun Nanofiber Membranes for Gravity-Driven Oil/Water Separation

**DOI:** 10.3390/membranes12040382

**Published:** 2022-03-31

**Authors:** Aatif Ali Shah, Youngmin Yoo, Ahrumi Park, Young Hoon Cho, You-In Park, Hosik Park

**Affiliations:** 1Green Carbon Research Center, Chemical Process Division, Korea Research Institute of Chemical Technology, Daejeon 34114, Korea; atifali.naqvi13@gmail.com (A.A.S.); ymyoo@krict.re.kr (Y.Y.); ptsdhd@krict.re.kr (A.P.); yhcho@krict.re.kr (Y.H.C.); yipark@krict.re.kr (Y.-I.P.); 2Department of Green Chemistry and Environmental Biotechnology, University of Science and Technology (UST), 217 Gajeong-ro, Yuseong-gu, Daejeon 34113, Korea

**Keywords:** poly(ethylene-co-vinyl alcohol), electrospinning, nanofiber, membrane, oil/water separation

## Abstract

Fabrication of highly efficient oil/water separation membranes is attractive and challenging work for the actual application of the membranes in the treatment of oily wastewater and cleaning up oil spills/oil leakage accidents. In this study, hydrophilic poly(ethylene-co-polyvinyl alcohol) (EVOH) nanofiber membranes were made using an electrospinning technique for oil/water separation. The as-prepared EVOH electrospun nanofiber membranes (ENMs) exhibited a super-hydrophilic property (water contact angle 33.74°) without further treatment. As prepared, ENMs can provide continuous separation of surfactant-free and surfactant-stabilized water-in-oil emulsions with high efficiency (i.e., flux 8200 L m^−2^ h^−1^ (LMH), separation efficiency: >99.9%). In addition, their high stability (i.e., reusable, mechanically robust) would broaden the conditions under which they can be employed in the real field oil/water separation applications. Various characterization techniques (including morphology investigation, pore size, porosity, mechanical properties, and performance test) for gravity-driven oil/water separation were employed to evaluate the newly prepared EVOH ENMs.

## 1. Introduction

Oil–water separation emerged as an inevitable challenge in the treatment of the oily wastewater from oil-spill clean-up and massive volumes of oily substances produced by various industries (such as petrochemical, mining, pharmaceutical, textile, metallurgical, and food industries) with industrial growth [1,2]. Moreover, the oil spill/oil leakage accidents that happen frequently during oil transportation and production have huge impacts on ocean life. It is also a waste of natural resources that shows the importance of developing better oil–water separation methods [3]. Another major source of oily water is petroleum wells, which often produce oil, gas, and water. It is always necessary to separate these valuable commodities for marketing and a safe environment [4,5,6]. Many efforts have been made to address this global concern, and several methods involving materials used for the purification of oily wastewater have been utilized, including conventional physical and chemical methods [7]. These conventional methods, including mechanical skimmers, adsorption, evaporation, and photocatalytic treatments, each have their limitations (such as energy, high pressure, high cost, use of toxic chemicals, and generation of secondary pollutants that narrow their practical applications [8,9,10]. Moreover, porous materials such as sponges, foams, and textiles have been utilized to absorb oils from water during oil leak/spill disasters, but these materials have low capacities and also absorb water simultaneously. In addition, recycling these materials with absorbed oil is difficult and time-consuming, which also limits their practical application [11]. Despite a variety of oil–water separation processes that have already been utilized, efforts continue to develop novel materials to meet the demand for better practical application [12]. The ideal process or material should have some specific characteristics, which include such as being environmentally friendly, inexpensive, and reusable materials with high efficiency, high capacity, and a high flux rate [1,13].

Keeping these ideal characteristics in view, membrane separation processes have emerged as promising technologies in the field of oil–water separation due to their high oil removal efficiency, low energy consumption, and compact design relative to conventional processes [14,15,16]. Another important factor that has played a significant role in membrane-based oil–water separation processes is the wettability of membranes. It is a common natural phenomenon, and membranes are generally either hydrophobic or hydrophilic depending on the characteristics of the membrane material [17]. Moreover, membrane wettability is not defined in terms of oil but rather is classified using water interactions with that specific surface. Membrane surfaces with water contact angles greater than 90° are called hydrophobic or oleophilic in nature, whereas those with angles less than 90° are called hydrophilic [18]. In addition, both kinds of membranes are employed in energy-intensive processes and are less considered for gravity-based oil–water separation. Because water always settles below the oil due to its greater density, a layer forms that is resistant to oil permeation [19,20,21]. On the other hand, hydrophilic membranes mostly have been used for gravity-driven oil/water separation processes and have higher resistance to the fouling phenomenon [22].

In membrane-based oil–water separation processes, membranes act as interfaces for the separation of either (but only one) component of the mixture (i.e., oil or water) as it moves from one side to the other. This is determined by the physical characteristics of the membranes, which include interfacial effects, porosity, and external parameters such as pressure. The surface porosity is a critical factor that greatly affects permeation (movement of water or oil) through the membrane. In addition, high separation efficiency or selectivity for oil/water from mixtures has been achieved by changing the surface characteristics (such as wettability and porosity) of a membrane with different hydrophobic/hydrophilic modifications [23,24,25]. Consequently, membrane separation technology has received great attention, and various membranes processes have been tried in this context, in particular, microfiltration (MF), ultrafiltration (UF), and reverse osmosis (RO) [26,27]. Over the years, a variety of membrane materials have been utilized including polyvinylidene difluoride (PVDF), polyvinyl alcohol (PVA), polyacrylonitrile (PAN), cellulose acetate (CA), cellulose triacetate (CTA), and polysulfone (PSF). Their success has depended on outstanding qualities such as mechanical strength, chemical resistance, and thermal and hydraulic stability, which make them suitable for oil–water separation [28,29]. It is a fact that there is no ideal polymer with only advantages for any particular application of oil–water separation. This includes the hydrophobic polymer-based membranes employed for energy-intensive processes. Similarly, hydrophilic polymers have issues with swelling and their mechanical properties. These problems have been addressed using different nanomaterials, and many modifications have been implemented to enhance the separation efficiency of membranes such as halloysite (HNT), titanium dioxide nanoparticles (NPs), graphene oxide (GO), silica NPs, sodium hydroxide, Zeolite, MWCNTs, ZrO_2_, ZnO, and silver NPs [30,31,32,33]. Although that porosity and pore size are critical factors that determine the performance of membranes for oil–water separation processes, the most widely utilized membrane fabrication method (phase-inversion process) results in smaller pore size and low porosity, which reduces membrane performance [34]. Pan et al. developed an anti-fouling TiO_2_ nanowire membrane for oil/water separation to improve the wettability and pore size of the membrane that synergistically improved membrane oil–water separation performance [35]. The results indicate that these membranes show high separation efficiency and outstanding anti-fouling performances after long-term oil/water separation. Qian et al. [36] presented ~100% oil/water separation efficiency by using TiO_2_/Ag nanoparticles/sulfonated graphene oxide-coated super hydrophilic copper meshes. The membrane exhibited excellent stability and durability, as well as a high separation efficiency after 10 consecutive cycles. Lin et al. [37] fabricated a self-cleaning MXene composite membrane with enhanced oil–water separation performance. The results showed that the composite membrane had a nearly 100% rejection ratio for oil/water emulsions from wastewater; more importantly, the composite membrane had excellent recyclability and self-cleaning ability after several cycling tests, which is the solution to the fouling problem in the oil–water separation process.

Electrospinning is a unique technique used to develop electrospun nanofibers with diameters ranging from micrometers to a few nanometers. Electrospun nanofiber membranes (ENMs) have attracted much attention due to exceptional features such as tunable porosity, high surface-area-to-volume ratio, good mechanical properties, tunable morphology, low cost, easy fiber functionalization, and good water permeability [38,39,40]. These features enhance performance and significantly reduce the energy consumption needed for the oil–water separation process [41].

In this study, polyethylene-co-polyvinyl alcohol (EVOH) polymer was used to develop ENMs for gravity-driven oil–water separation. Three types of ethylene vinyl alcohol (EVOH) had different contents of ethylene. These polymers were electrospun and one of the best among them was selected based on its resulting EN hydrophilicity and feasibility for electrospinning. To the best of our knowledge, this is first work on EVOH ENM for oil–water application. The novelty of this work is the development of pristine EVOH ENMs with outstanding characteristics such as no energy required (gravity-driven), initially hydrophilic (no need for modification or incorporation of nanomaterials), mechanically robust, non-fouling, environmentally benign, biodegradable, and significantly higher performance than those reported for other membranes. The main highlight of this work is that a durable and high-performance membrane was fabricated via a one-step electrospinning process and used as is for the oil–water separation process without any other post-treatment. These ENMs exhibit high porosity and performance under gravity using different oils. These include volatile hydrocarbons (i.e., hexane, toluene, and cyclohexane), fixed oils (i.e., canola, soybean, and olive oil)/water, and hexane/water emulsion.

## 2. Materials and Methods

### 2.1. Materials

In this work, all chemicals and reagents used were of analytical grade and were used as received. Poly(ethylene-co-polyvinyl alcohol) with different ethylene contents (i.e., EVOH with 44% and 27% ethylene contents, denoted as 44EVOH and 27EVOH, respectively) and dimethyl sulfoxide (DMSO) were purchased from Sigma Aldrich and were used to prepare the electrospun fibrous membranes. Sodium dodecyl sulfate (SDS) surfactant was purchased from Sigma Aldrich to prepare the hexane–water emulsion. In addition, the commercial volatile hydrocarbons hexane, toluene, and cyclohexane were bought from Sigma Aldrich. The fixed oils (i.e., canola, soybean, and olive oil) were purchased from a local market in South Korea. Deionized (DI) water with a resistivity of 18.2 MΩ·cm obtained from a deionized water purification system (RiOs™ Essential, Millipore, Temecula, CA, USA) was used for the separation experiments.

### 2.2. Preparation of EVOH Electrospun Nanofiber Membranes

First, 44EVOH was utilized to fabricate electrospun nanofiber membranes using solutions with different concentrations (i.e., 20, 22, and 24 wt%) to determine the optimal EVOH concentration for fabrication of the final membrane. These doping solutions were prepared by adding 20, 22, and 24 wt% of 44EVOH polymer in DMSO solvent and stirring for 24 h at 60 °C to obtain a homogeneous solution for the electrospinning process. This process was conducted using a laboratory-scale apparatus comprising three major components: a high voltage power supply, a spinneret (syringe with pipette tip), and a grounded collector (a rotating stainless-steel cylinder covered with a sheet of aluminum foil). As a result, the optimized polymer concentration (22 wt% of 44EVOH) was determined after various characterizations. Then, the operating conditions for the electrospinning process were optimized by controlling the distance of the needle from the drum collector, as well as the flow rate, by applying a constant voltage of 12 kV for 44EVOH and 27EVOH. Finally, the 27EVOH ENMs prepared under optimal electrospinning conditions (i.e., distance = 14 cm, flow rate = 0.5 mL/h) were characterized and their separation performance evaluated.

### 2.3. Separation Setup

For the oil/water separation experiments, a separation setup was designed for a batch process. Figure 1 presents a schematic diagram of oil/water separation process. Because an EVOH electrospun nanofiber membrane is hydrophilic, the water fraction permeates through the membrane while the oils are retained at the membrane surface under gravity. The process was run until only pure water remained on the permeate side and the oil was on the feed side.

### 2.4. Characterization

The surface morphology of the membrane was studied using a field-emission scanning electron microscope (FE-SEM, Mira 3, Tescan, Czech Republic). The fiber diameter distribution of the electrospun nanofiber was examined using ImageJ software. The surface wettability of the EVOH electrospun nanofiber membranes was investigated by measuring the water contact angle (WCA, drop shape analyzer KRUSS GmbH 22,453 Hamburg, Germany) using the sessile drop method. The pore size of the prepared membranes was analyzed using a PMI Capillary Flow Porometer CFP 1500A (Porous Materials, Inc. NY, USA). Samples were first wet with Galwick surface tension of 15.9 dynes/cm and then the liquid was displaced from the pores using pressurized gas. The instrument measures the pore size from the gas pressure required to displace liquid from the pores. The mechanical properties of the optimized EVOH electrospun nanofiber mat were examined using an Instron Corporation Automated Materials Testing system at room temperature. The separation performance in terms of water flux was measured using a lab-scale separation setup, as shown in Figure 1. The water flux ($J_{w}$, LMH) was measured by calculating the water permeation volume per unit time for the specific area (0.00113 m^2^), using the following equation.
(1)$$J_{w}=\frac{V}{A\cdot \Delta t}$$

The separation efficiency was verified using a Raman Spectroscopy (Gloucestershire, United Kingdom) and an HP 8453 UV-Visible Spectroscopy System, and the obtained spectra were analyzed using the HP ChemiStation software 5890 series. The porosity of the EVOH electrospun nanofiber membranes was calculated using the dry–wet weight method and the following equation.
(2)$$P_{r}=\frac{m_{w-}m_{d}}{\rho \cdot S\cdot \;L}$$
where $m_{w}$ is the wet membrane weight (g), $m_{d}$ is the dry membrane weight (g), ρ is density of DI water (0.998 g/mL), S is the effective membrane area, and L is the membrane thickness. The flux decline ratio was calculated using the following equation.
(3)$$Decline\;ratio=\frac{J_{w,i-}J_{w,\;f}}{J_{w,i\;}}\times 100$$

Here, $J_{w,i}$ and $J_{w,\;f}$ are the original flux value and new flux value of the next cycle, respectively. An optical microscope (Nikon H600, Nikon Instrument Inc., New York, NY, USA) was utilized to observe the emulsion before and after the separation test.

## 3. Results and Discussion

The electrospinning solution was prepared simply by mixing EVOH into the DMSO. Details of the fabrication process of the EVOH ENMs are presented in the experimental section and are shown in Figure 2. The ENMs made with polymer at different concentrations were prepared with the electrospinning conditions kept constant, as shown in Table 1.

### 3.1. Optimization of the Electrospinning Conditions

The structure morphologies of the 44EVOH ENMs prepared at different concentrations were examined using FE-SEM. Figure 3 shows SEM images of the surface and the fiber diameter distribution of the newly fabricated ENMs. The results show that at lower polymer concentrations, beads appeared along with fibers. It is a fact that the viscoelasticity of the solution, the charge density carried by the jet, and the surface tension of the solution are the main factors that influence the formation of beaded fibers.

At lower viscosity and inadequate charge density, the jet was unable to cause suitable elongation of fibers because the charge density was insufficient to overcome the high surface tension. As a result, beads formed [42]. In the electrospinning process, it is important to have suitable threshold conditions to overcome the viscosity and surface tension adequate to form smooth nanofibers. When the polymer concentration was increased to 22 wt%, smooth fibers formed with no bead formation. A further increase in the polymer concentration resulted in thick, wrinkled nanofibers [43].

On the other hand, the fiber diameter distribution demonstrated a direct relation with polymer concentration, such as when an increase in the concentration resulted in increased nanofiber diameter. This happened because of the increase in the viscosity of the polymer solution and an increase in the extent of polymer chains in the solution [44]. The water contact angle of the fabricated membranes confirmed the hydrophilic nature of 44EVOH and with the increase in the polymer concentration, there was a relative decrease in the contact angle due to the high fiber diameter and pore size. Figure 4 shows that 44EVOH_22% exhibited a lower contact angle (33.74°) than 44EVOH_20% and 44EVOH_24%. The 44EVOH_24% contact angle (42.2°) indicated a different story and this is because, at higher concentrations, the fibers melt and form a film. Thus, although the fiber diameter increases with the polymer concentration, the film formation and non-uniformity significantly influence the surface wettability. These results are consistent with those in the literature except that the water contact angle at a higher concentration was slightly higher due to its morphology. Thus, 44EVOH_22% was chosen as the optimum polymer concentration for further experiments due to its perfect morphology and hydrophilicity.

After optimization of the polymer concentration (i.e., 44EVOH_22%), it was necessary to optimize the electrospinning conditions with the optimum polymer concentration because these conditions have a strong influence on the resulting electrospun nanofibers (affecting the structure, fiber diameter, pore size, and porosity). It is a fact that the water permeation of EVOH with 44 mol% ethylene (44EVOH) is approximately eight times lower than EVOH with 27 mol% (27EVOH). The reason for this is that EVOH with lower polyethylene (PE) content behaves similar to polyvinyl alcohol (PVOH), which has the ability to attract more water due to its functional group [45,46]. Therefore, we chose 27EVOH instead of 44EVOH to optimize the electrospinning conditions. Figure 5 presents the surface morphology and fiber diameter distribution of the newly prepared 27EVOH ENMs at different flow rates and distances. With an increase in the flow rate, the fiber diameter increased because there was more volume for electro-spinning when other electrospinning conditions, such as the needle gauge, were kept constant. The fiber density was high, fibers thick, and distribution very random due to the short distance and inadequate elongation of the fibers at the distance of 10 cm. Moreover, the greater volume to electro-spin/time under the same conditions and distance was insufficient to complete evaporation at the higher flow rate of 0.5 mL/h. This is why the fibers melted down and started to form a film. Another explanation for this at a high flow rate is that residual solvent remained in the deposited fibers because there was not enough time for evaporation. This may cause the fibers to fuse [47,48].

The effect of the distance from the needle to the drum collector on the fiber morphology can be seen in the columns of images. In the first column, with increased distance, the fibers became smoother, the fiber diameter increased, and the nanofiber density decreased. This can be explained by an increase in the nanofiber elongation at the longer distance. This is why when the nanofiber diameter decreased, the fiber jet stability was decreased slightly and the fibers spread on the collector widely due to the weak electric field [49]. Nanofibers became smooth and uniform even at the highest flow rate of 0.5 mL/h at the longer distance (i.e., 14 cm) because there was enough time for evaporation so that the fiber melting effect disappeared. For oil/water separation, the porosity and pore size of membranes are very important and influence the separation performance. A high porosity, pore size, and high permeation rate of water in a hydrophilic membrane are critical factors during an oil/water separation experiment. It is a fact that with increases in the flow rate, the nanofiber diameter increases and consequently, the pore size and porosity of the resulting nanofiber mats increase. In this study, similar trends were seen: with an increase in the nanofiber diameter, the porosity and pore size increased, along with an increase in the flow rate. These results are consistent with the FE-SEM images in Figure 5 [50,51,52].

On the other hand, with the change in the distance between the needle and drum collector (i.e., 10, 12, and 14 cm) while keeping the flow rate constant, it was observed that the nanofiber diameter decreased due to a greater elongation time, but the high porosity and pore size contradicted the above explanation. This can be explained as follows: even though the fiber diameter decreased and pore size should have decreased, Figure 5 exhibits fewer fibers at a greater distance due to instability of the jet (i.e., weak electric field), resulting in a porous ENM. Considering these factors, 14 cm and 0.5 mL/h were selected as the optimal conditions for further separation experiments. These membranes were further characterized in terms of porosity, pore size, and water flux (i.e., for a two-phase hexane–water system presented in Figure 6). The hexane–water solution was prepared as 20 mL of hexane and methylene blue dye 80 mL aqueous solution (as colorized water) for measuring the water permeate flow rate. Five cycles were conducted for each membrane, and the average of the fluxes was calculated. Figure 6a–c shows the flux results of all experimental membranes corresponding to electrospinning conditions. These results suggested that the EVOH membrane with 0.5 mL/h and distance of 14 cm had the highest water flux in the two-phase system. Figure 6d illustrates the complete process for separation of the parts of the hexane–water system at different time intervals. For this batch process, separation was completed in 120 s. These results are consistent with the characterization of these membranes as mentioned above. These optimized conditions (i.e., 0.5 mL/h and distance 14 cm) were utilized to develop membranes for oil/water separation experiments using different oils and emulsion. Apparently, all membranes prepared with a distance from the needle to the drum collector of 14 cm seemed suitable, with homogeneous fibers. Fiber density and thickness were higher at a lower flow rate, indicating a smaller pore size and greater water permeation resistance.

### 3.2. Mechanical Property Analysis

In general, the mechanical properties of the electrospun nanofiber membranes are very important, and it has been a concern to use ENMs in different applications. As shown in Figure 7, the mechanical property (~9.03 ± 1.09 MPa) of the optimized 27EVOH_22% electrospun nanofiber-based membrane was good enough to use in oil/water separation without any support. This is a significant improvement in the modulus associated with the increase in the nanofiber diameter caused by a high flow rate. Moreover, in this study, a gravity-based oil/water separation experiment was conducted. Even though the strong mechanical property was less considered in this study, the mechanical properties of the ENMs were higher than or comparable to the values in the literature [29]. Ma et al. developed electrospun nanofibers for oil/water separation, and the maximum tensile strength of their mat was approximately ~60–65 kPa [41]. Similarly, Ahmed et al. prepared cellulose/PVDF-HFP composite ENMs for efficient oil/water separation and their stress–strain curve for pure and composite ENMs showed a tensile strength from 5.5 to 8.6 MPa [53]. Another advantage of EVOH is that it provides electrospun nanofiber mats with strong mechanical properties without modification and, if directly employed for repeated separation experiments, results in higher performance than modified membranes based on the literature [54].

### 3.3. Membrane Performance Evaluation

#### 3.3.1. Separation Efficiency of Different Oil-Water Systems

To test the separation capability of the EVOH membranes, a series of two-phase oil–water systems was achieved using the separation setup shown in Figure 1. The high porosity, good wettability, and satisfactory mechanical strength of the as-spun membrane (EVOH ENM) allowed the membrane to separate the oil–water mixture efficiently. The characteristic of high porosity coupled with hydrophilicity makes the EVOH ENM a promising membrane for oil−water separation. Another important factor that plays important role in oil–water separation performance is the polar–polar interaction via van der Waals force between the abundance of hydrophilic functional groups on the EVOH ENM surface and water. These systems included hexane, toluene, cyclohexane, canola, soybean, and olive oil; that is, surfactant-free water-in-oil solutions with a droplet size at the micrometer scale. The as-prepared two-phase oil–water solutions were poured onto EVOH membranes. Water immediately permeated the membrane, while the oil droplets rebounded once touching the EVOH membrane. All oils were retained above the membrane. The whole process was driven completely by gravity without any other external force. All of the oil–water solutions were separated successfully in one step. Figure 8a presents the separation efficiency of the 27EVOH_22% membrane (>99.95% and even up to 99.999%) and shows extraordinary separation efficiency. These results were verified using Vis–UV spectroscopy and Ramon spectroscopy of pure oil samples and permeate. The spectroscopy results in Figure 8c,d for pure oils and permeates exhibited no signs of oil on the permeate side. Moreover, all permeate samples showed spectra of pure water. To confirm these results further, Ramen spectroscopy (Figure 8e,f) was also conducted for pure oils and permeates. The results confirmed a very high separation efficiency of ~99.99%: there was no trace of oil on the permeate side.

On the other hand, the water fluxes were measured in these experiments (obtained by calculating the permeate volume per unit time from the specific membrane area) as shown in Figure 8b. All samples exhibited high water fluxes, such as 8200.3 ± 103.2, 7827.6 ± 83.3, 8042.1 ± 86.7, 7774.6 ± 89.9, 7667.1 ± 101.2, and 7561.9 ± 125.9 L m^−2^ h^−1^. These values are for surfactant-free two-phase processes with water-hexane, water-toluene, water-cyclohexane, water-soybean, water-canola, and water-olive oil, respectively. We conducted oil/water separation with various oils, but there was no significant difference among them. Tai et al. prepared an electrospun carbon–silica nanofibrous membrane for gravity-driven oil/water separation. The oil–water separation results showed a maximum of flux 3032.4 ± 234.6 L m^−2^ h^−1^ for petroleum spirit, i.e., not even close to our membrane flux. Moreover, they modified the nanofibers with nanomaterial to improve the membrane properties [54]. To the best of our knowledge, in this work, the flux obtained using the EVOH membrane and gravity method is unprecedented.

To test the flux recovery of the optimized membrane, a water–hexane two-phase system was employed, and 20 cycles were completed as shown in Figure 9a. After each cycle, the membrane was washed thoroughly using ethanol and DI water to recover the flux. It was observed that there was no sharp decline in water flux with each cycle but there was a gradual decrease. After the last (20th) cycle, the flux value was 7867.21 L m^−2^ h^−1^, which is still a high flux level. Rates of decline were measured for each cycle, and the overall decline rate for all 20 cycles was less than 2%. This shows that optimized EVOH membranes exhibit exceptional antifouling performance and also proves the long-term stability and viability for practical application of the membrane. This is due to the hydrophilic nature of EVOH, usually with enhanced hydrophilicity of the membrane surface that increased the resistance to membrane fouling because many foulants are hydrophobic in nature.

#### 3.3.2. Oil-Water Emulsion Performance

To determine the separation capability of optimized EVOH electrospun nanofiber membranes (i.e., 27EVOH_22%) for oil–water emulsion, a hexane-in-water emulsion with sodium dodecyl sulfate (SDS) was prepared. Then, the emulsion was poured into a filtration cell to perform separation under gravity. It was noticed that 27EVOH_22% membranes exhibited high separation efficiencies (>99.96%) for both surfactant-free and surfactant-stabilized emulsions. This experiment was repeated for 15 cycles, and after each cycle, the membrane was rinsed thoroughly with DI water and ethanol. Figure 9b illustrates the permeation of hexane-in-water emulsion for the 27EVOH_22% membrane: and the amount of filtrate decreased with increasing cycles, and little water permeated through the membrane compared with the 1st cycle; the membrane with weak hydrophilicity was easily fouled, and the pores were quickly clogged, indicating that the hydrophilic EVOH EN membrane has poor antifouling ability against oil-in-water emulsion stabilized by anionic surfactants. This can be explained as the fouling of surfactant on the membrane surface may be due to the non-specific interactions between the surfactant molecules and the membrane surface such as van der Waals interactions, hydrophobic interactions, hydrogen bonding, and electrostatic interactions [55].

On the other hand, the 27EVOH_22% membrane showed excellent separation ability with oil rejection >99%, and this result was confirmed with optical microscopy of an oil-in-water emulsion before and after separation, as shown in Figure 9c. It was evident from the optical photomicrographs that oil droplets were uniformly distributed in the emulsion prior to the separation experiment; however, no oil droplet was seen in the filtrate image. The stability of membranes in harsh environments is an important factor for practical applications. EVOH membranes have the ability to persist in such situations and exhibit excellent antifouling characteristics without any modifications because of their hydrophilic nature.

An overall oil/water separation performance comparison between the membrane prepared in this work and reported in the literature is summarized in Table 2, which clearly shows that EVOH ENM has outstanding performance, considering the permeation flux. The pore size is a critical parameter controlling the flux and separation efficiency. Table 2 presents a pore size comparison of the EVOH EN membrane with reported data, and EVOH ENM has a pore diameter of ~3.2 µm, which is comparatively larger than reported ones, which justifies the higher flux in this work other than hydrophilicity. The separation flux was very stable even after the 20th cycle, and the membranes could be used in another round of cycles in the two-phase oil–water system. In addition, the flux of EVOH ENM was comparable to or even higher than most of the modified reported membranes in the literature. EVOH ENM was prepared by a single-step electrospinning process without any post-treatment but its performance was exceptional. These results demonstrate that EVOH ENMs can be used as filters for oil/water separation.

## 4. Conclusions

In this experimental work, super hydrophilic EVOH electrospun nanofiber membranes (ENMs) were successfully fabricated using an inexpensive electrospinning technique. These ENMs showed good surface hydrophilicity (water contact angle 33.74°). The mechanical properties of ENMs are always a major concern, but these EVOH nanofibers presented good tensile strength (9.03 MPa), which is more than enough because the separation experiment was carried out under gravity. The separation efficiency and water flux were exceptionally good (>99.9% and 8200.3 L m^−2^ h^−1^, respectively) under gravity. Moreover, the EVOH electrospun nanofiber demonstrated high flux recovery and long-term stability without modification. EVOH has some exceptional characteristics that make it more viable for oil–water separation applications. These include biodegradability, stability in harsh environments, and super hydrophilicity. Furthermore, the present investigation also showed that EVOH ENMs have potential uses as high-efficiency liquid-separation membranes for the separation of emulsified oil–water solutions. EVOH electrospun nanofiber membranes exhibited good flux recovery, long-term stability, good mechanical properties, and high separation efficiency, which are necessary for real emulsion treatment on a mass scale. The EVOH polymer is promising for various applications that include the separation of emulsified wastewater, fuel treatment, and even crude oil treatment. In conclusion, electrospinning is a low-cost, scalable fiber fabrication technique that could be used to produce porous, mechanically robust nanofiber membranes with controllable morphology for oil–water separations.

## Figures and Tables

**Figure 1 membranes-12-00382-f001:**
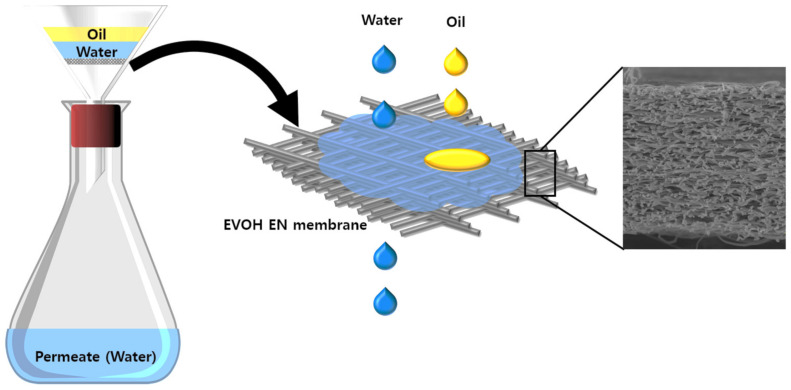
Schematic diagram of the oil/water separation process.

**Figure 2 membranes-12-00382-f002:**
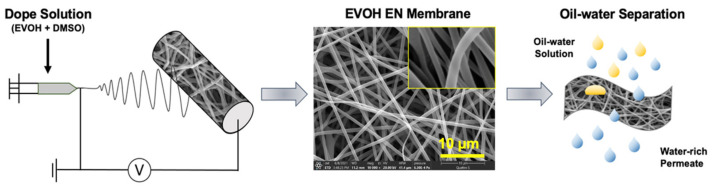
Fabrication of the hydrophilic EVOH nanofibrous membrane and its application for oil–water separation.

**Figure 3 membranes-12-00382-f003:**
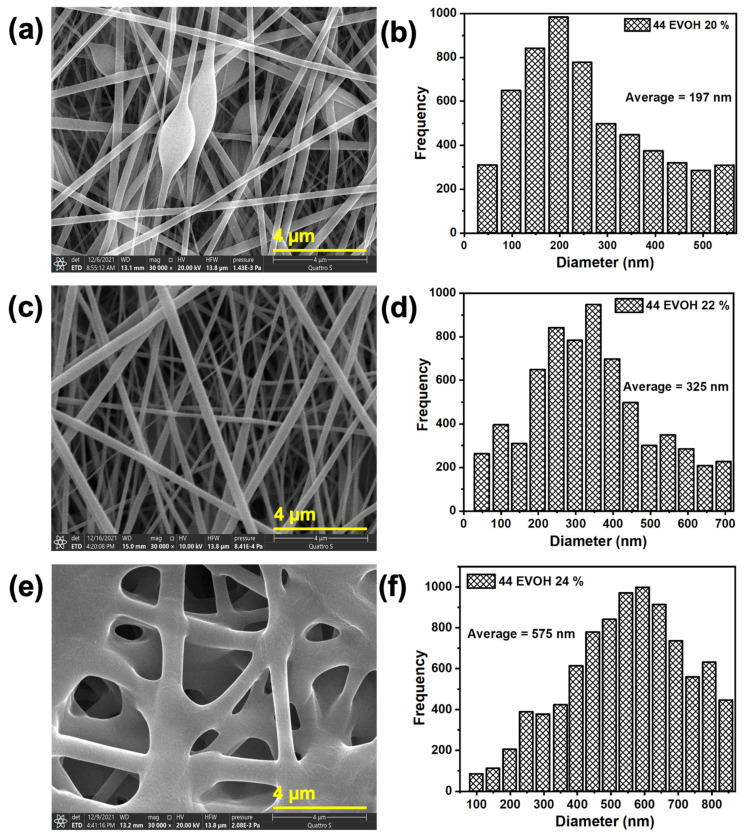
SEM images and fiber diameter of 44EVOH at different polymer concentrations: (**a**,**b**) 20 wt%, (**c**,**d**) 22 wt%, and (**e**,**f**) 24 wt%.

**Figure 4 membranes-12-00382-f004:**
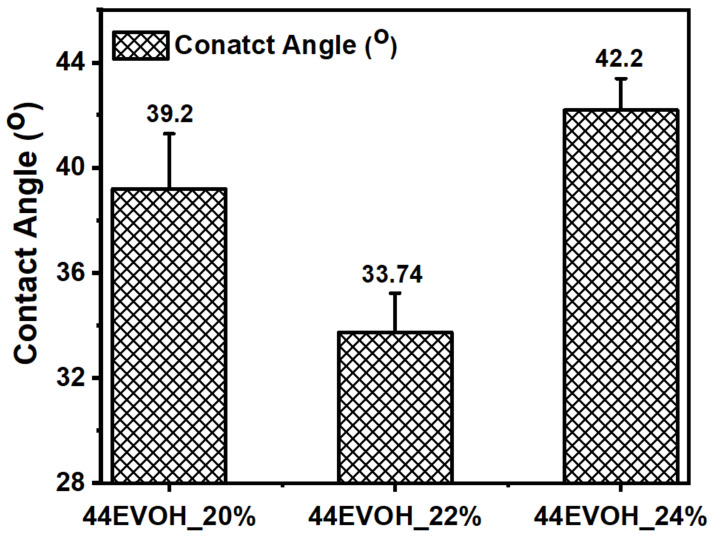
Water contact angle of the 44EVOH nanofibers corresponding to the respective polymer concentration.

**Figure 5 membranes-12-00382-f005:**
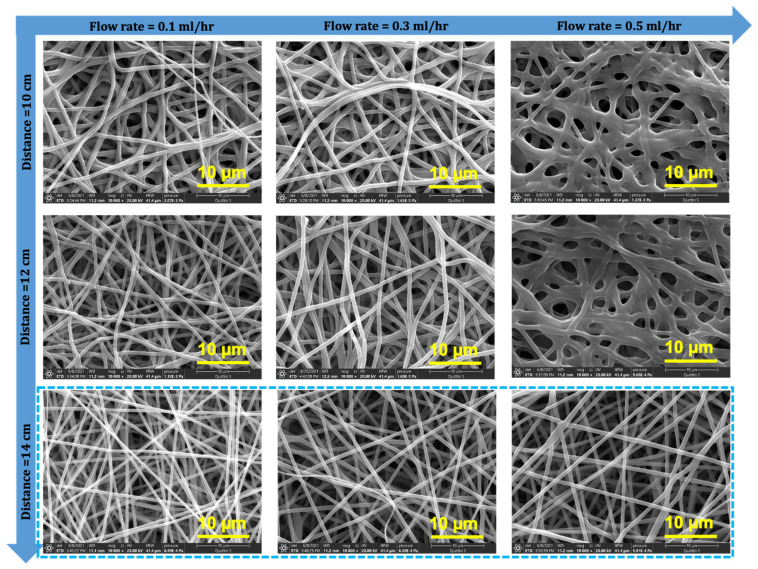
FE-SEM images of 27EVOH_22% at different flow rates and distances used to optimize and study the effect of the electrospinning condition on nanofiber morphology.

**Figure 6 membranes-12-00382-f006:**
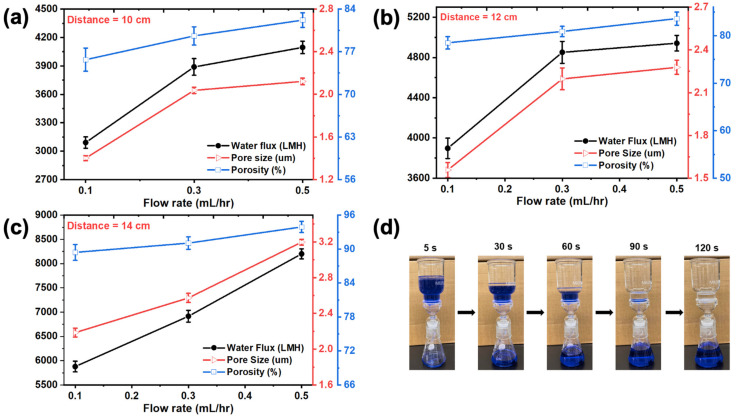
27EVOH_22% ENM properties at different flow rates and distances between the collector and needle: (**a**) 10 cm, (**b**) 12 cm, and (**c**) 14 cm. (**d**) Visuals of the separation process completed in 120 s.

**Figure 7 membranes-12-00382-f007:**
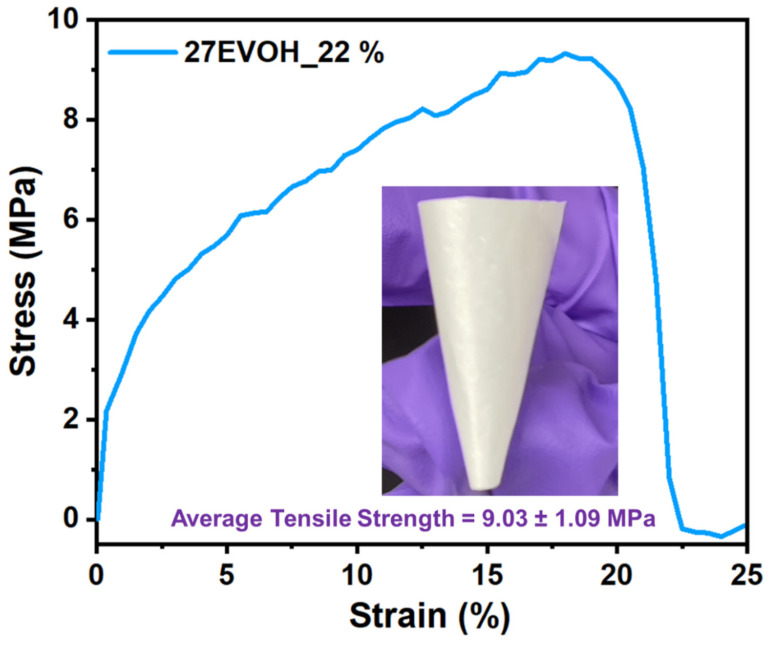
Stress–strain curve of the 27EVOH_22% membrane. The inset is a photograph of the bent electrospun nanofibers.

**Figure 8 membranes-12-00382-f008:**
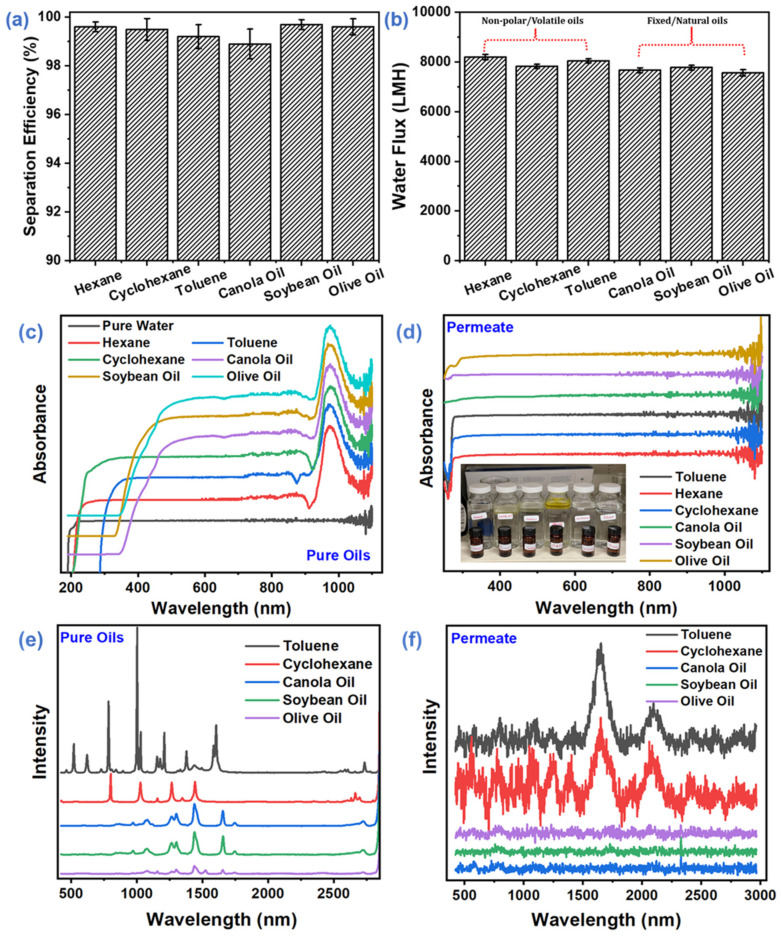
Separation performance of the optimized 27EVOH_22% ENMs for different oil–water mixtures: (**a**) Separation efficiency, (**b**) Water flux and UV–visible spectra of pure oils and permeate, (**c**) Pure oils, (**d**) Permeates. Raman spectroscopy spectra for pure oils and permeates, (**e**) Pure oils, and (**f**) Permeates to confirm separation efficiency >99.9%.

**Figure 9 membranes-12-00382-f009:**
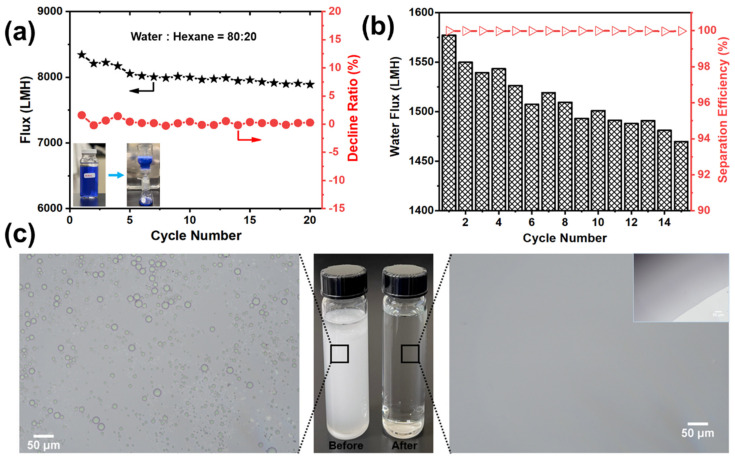
(**a**) Flux recovery and ratio of decline of the EVOH membrane with increasing cycle number. Oil–water emulsion separation results of the 27EVOH_22% membrane: (**b**) Flux and separation efficiency of emulsion through the membrane and (**c**) Optical microscopy images of the emulsion before and after separation. The inset is an image of a permeate water drop near the boundary line.

**Table 1 membranes-12-00382-t001:** Composition of the spinning solution and the electrospinning conditions.

Dope Solution for Electrospinning			Electrospinning Conditions	
ID	Polymer (wt%)	DMSO (wt%)	Parameter	Number
44EVOH_20%	20	80	Voltage (kV)	12
44EVOH_22%	22	78	Distance (cm)	12
44EVOH_24%	24	76	Flow rate (mL/h)	0.1

**Table 2 membranes-12-00382-t002:** Oil–water separation performance comparison between EVOH ENM and literature reports.

Membrane Material	Method	Oil-Water System	Flux (L m^−2^ h^−1^)	Pore Size (µm)	Cycle Numbers	Reference
EVOH ENM	Gravity	Water/ n-hexane	8200.3 ± 103.2	3.2 ± 0.032	20	This work
MFS/CC-DKGM	Filtration	Water/ n-hexane	4702	-	20	[56]
Cellulose fiber/LDH	Gravity	Water/ Chloroform	4968	~0.15	50	[57]
CU mesh/ Cement	Gravity	Water/ n-hexane	5000	-	1	[58]
TiO_2_-PVDF	Vacuum F.	Water/ n-hexane	785	-	3	[59]
PDMS glass filter	Gravity	Water/ Chloroform	6351	1.6	11	[60]
Al_2_O_3_ membranes	Gravity	Water/Octane	14.1 L m^−2^ s^−1^ kPa^−1^	15~30	1	[61]
PDMS-PI membrane	Gravity	Water/DCM	4443.2 ± 70	-	20	[62]
BN-CuSA_2_ membrane	Gravity	Water/DCM	1667.63	0.00306	10	[63]
NM88B@QFM	Gravity	Water/ n-hexane	489	1.148	15	[64]
CNTs@PVDF	Gravity	Water/ Chloroform	6600	0.0124	15	[65]

## Data Availability

The data available in this study are available on request from the corresponding author.

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
