# Peer review of "Poly(ethylene-co-vinyl alcohol) Electrospun Nanofiber Membranes for Gravity-Driven Oil/Water Separation"

_membranes, 2022, doi:10.3390/membranes12040382_

Round 1
Reviewer 1 Report
This article is more interesting and has done a lot of experimental work. But some issues require reasonable explanations from the author:
1.In our traditional oil-water separation process, the best way to separate water from oil is to prepare a lipophilic and hydrophobic fiber membrane, but the article says it should be hydrophilic (water contact angle 33.74°), It is bound to cause incomplete separation of oil and water. How to explain this problem.
2. The schematic representation in Figure 1 is not accurate. First, it is not a monolayer, and second, are oil and water separated by the size of the pores? In other words, water must adhere to the surface of the fiber and form condensed water, so I have doubts about your experimental results.
3. The superposition of the SEM image and the histogram in Figure 3 makes it unclear. Please redesign the image.
4. The article only pays attention to the elaboration of the experimental results, and the analysis of the influencing factors and principles of gravity filtration is less, and further analysis and increase are needed. Such as fiber fineness, layer thickness, porosity, the concentration of oil-water mixture, the change of liquid gravitational potential energy and other factors.
5. The language needs to be polished, the key performance analysis parts need to be highlighted, and the subtitles need to be concise (3.1 and 3.2 are similar).
Reviewer 2 Report
The state of art on gravity driven oil water separation is not presented well. The novelty is not well described considering the state of the art. Previous studies are not cited. Results are not well discussed.
Reviewer 3 Report
The hydrophilic poly(ethylene-co-polyvinyl alcohol) (EVOH) nanofiber membranes were fabricated using an electrospinning technique for oil/water separation. The prepared membranes provide separation of both surfactant-free and surfactant-stabilized water-in-oil emulsions. 1. In the part of introduction, it is suggested to add the theoretical basis and innovation of the development of modified membrane. 2. What is the performance level compared with the literatures? 3. The decline of membrane flux for membrane recycling is caused by membrane fouling. What are the main fouling components deposited on membrane surface? Will there be negative effects in the treatment of actual complicated water? 4. How do the physical and chemical properties of the prepared membrane, such as membrane porosity, affect the oil-water separation performance?Author Response
Please see the attachment.

Round 2
Reviewer 2 Report
The authors have improved the manuscript compared to original version however it is still confusing and it suffer from lack of discussion. Even the authors provide comparison with previous reports, no details such as cycle number and physical properties of the membranes are provided for better comparison. Considering electrospun membranes are already reported, novelty should be better explained. The high flux should be explained comparing other membranes structure reported. Physical properties are not discussed sufficiently. Pictures are too small. More cycles are reported in previous reports. Fouling properties have not discussed. No pictures of the membranes after cycles. Physical properties and filtration properties and results are not discussed sufficiently.
Author Response
We are very thankful to the reviewer for the critical comments. We tried to address all the comments in our rebuttal.

Reviewer 3 Report
Accept
Author Response
We are very thankful to the reviewer, to agree with our first review response and agree to accept our manuscript for publication.

Round 3
Reviewer 2 Report
accept